# Features of the Metabolic Profile of Saliva in Lung Cancer and COPD: The Effect of Smoking Status

**DOI:** 10.3390/metabo11050289

**Published:** 2021-04-30

**Authors:** Lyudmila V. Bel’skaya, Elena A. Sarf, Denis V. Solomatin, Victor K. Kosenok

**Affiliations:** 1Biochemistry Research Laboratory, Omsk State Pedagogical University, 644099 Omsk, Russia; nemcha@mail.ru; 2Department of Mathematics and Mathematical Education, Omsk State Pedagogical University, 644099 Omsk, Russia; denis_2001j@bk.ru; 3Department of Oncology, Omsk State Medical University, 644099 Omsk, Russia; victorkosenok@gmail.com

**Keywords:** lung cancer, chronic obstructive pulmonary disease, smoking, saliva, biochemistry, diagnostics

## Abstract

The aim of the study was to compare the metabolic characteristics of the salivary composition in lung cancer, chronic obstructive pulmonary disease (COPD) and their combination, depending on the smoking history. The study group included 392 patients with lung cancer of various histological types. The division into subgroups was carried out in accordance with the severity of COPD and smoking experience. Salivary biochemical composition was determined according to 34 indicators. For data processing, the principal component method was used. Different groups of biochemical saliva markers are informative when separately accounting for the smoking factor and the presence of COPD in lung cancer. For smoking, antioxidant enzymes and electrolyte components of saliva are informative; for COPD metabolic enzymes, lipid peroxidation products, sialic acids and electrolyte components are informative. While taking into account the smoking factor and the presence of COPD, biochemical markers corresponding to the presence/absence and severity of COPD are the priority. Changes occurring in the background of smoking are of a secondary nature, manifesting as much as possible with a smoking history of more than 50 pack-years. Thus, the metabolic changes that occur in lung cancer in combination with COPD, depending on the smoking factor, can be estimated using saliva.

## 1. Introduction

Lung cancer is the first cause of cancer death in the world, accounting for up to 13% of all cancer deaths with more than 1,400,000 deaths annually [1,2]. Chronic obstructive pulmonary disease (COPD) is a common co-morbidity in lung cancer, affecting up to 40% of lung cancer patients [3,4,5]. At the same time, in an independent variant, COPD is the fourth cause of death in the world with a current prevalence of about 10% [6]. It is known that the majority of patients with lung cancer who have COPD comorbidities are smokers [7], and the risk of developing lung cancer increases with the increase in smoking history [8,9,10]. Cigarette smoke contains more than 1000 oxidants/free radicals and reactive 4700 chemical compounds, including aldehydes, quinones, semiquinones, nitrosamines, benzo pyrene and other carcinogens, and it is a risk factor for development of COPD and lung cancer, causing chronic inflammation [11]. Repetitive lung damage and repair caused by chronic inflammation in COPD is thought to contribute to the development of lung cancer [12,13,14]. Thus, both smoking and COPD are factors in lung cancer development [15,16].

Saliva can be used to assess the metabolic changes that occur in lung cancer and COPD [17,18]. Saliva is an easily collected, non-invasive biological fluid for the diagnosis of diseases, including lung cancer [19]. It has been shown that the composition of saliva can determine lung cancer with a sensitivity and specificity of 90% [20,21,22,23,24,25,26,27,28]. This attention of researchers to the study of the composition of saliva in lung cancer confirms its potential applicability for diagnostic purposes.

Previously, we identified 34 biochemical parameters of saliva, which statistically significantly change in lung cancer of various histological types in comparison with healthy controls [17]. We have previously shown that the presence of COPD as a concomitant pathology does not fundamentally change the metabolic profile of saliva but increases the range of changes in the corresponding biochemical indicators [29]. Biochemical indicators (catalase, imidazole compounds, sialic acids, lactate dehydrogenase) have been identified, which can be used to monitor patients from risk groups for the timely diagnosis of lung cancer, in particular patients with early stages of COPD [29]. In this study, we compared the metabolic profiles of saliva within a group of patients with lung cancer depending on the presence or absence of COPD while taking into account the smoking status. The indicators of overall survival of patients with lung cancer and COPD were estimated depending on the smoking history. The aim of the study was to test the hypothesis that the presence of COPD has a greater effect on the metabolic profile of saliva and survival rates in lung cancer than smoking.

## 2. Results

### 2.1. Patient Characteristics

In the group of patients with lung cancer, the majority are non-smoking patients (219 people, 55.9%). Groups of smokers and non-smokers are not homogeneous in their composition. Thus, in the group of smokers, there are only seven women (8.0% of the total number of women), while among male patients 51.9% smoke. Correspondingly, 92.0% of women and 48.1% of men are non-smokers. It was noted that in the group of non-smokers, the predominant histological type of lung cancer is adenocarcinoma, while the ratio of the number of patients with adenocarcinoma and squamous cell carcinoma is 1.75. For the group of smokers, the proportions of adenocarcinoma and squamous cell carcinoma are comparable, and the ratio is 1.08. The proportion of neuroendocrine cancer in non-smokers is higher. It should be noted the predominance of the peripheral form of tumor growth over the central one for nonsmoking patients. For non-smokers, the ratio of the coefficient of peripheral growth to central growth was 2.40, while for smokers, it was 1.69.

The distribution of patients according to smoking status and the presence of COPD is shown in Table 1.

Obviously, the maximum proportion of nonsmoking patients falls in the group of lung cancer patients without COPD. At the same time, in the group COPD II, the proportion of smokers with long experience increases. Therefore, the group Smokers I+COPD II consists of only four people, while the group Smokers III+COPD II consists of 14 people. The age of patients with different smoking histories naturally increases. However, no statistically significant differences were found between groups with the same smoking experience by age.

### 2.2. The Salivary Biochemical Composition of Lung Cancer Patients Depending on Smoking

At the first stage, the study group was divided according to the smoking status (Figure 1A,B). Biochemical analysis of saliva showed that two parameters differ statistically significantly: chloride content and catalase activity (Appendix A). The chloride content in the group of non-smokers is significantly lower (−9.2%, *p* = 0.0163), while the catalase activity is higher (+10.3%, *p* = 0.0343). In addition to chlorides and catalase, the AST/ALT ratio and antioxidant activity (AOA) were selected as significant for PCA analysis (Figure 1B). It is shown that all of the listed indicators make approximately the same contribution to PC1 (Appendix A), while AST/ALT ratio and AOA make a greater contribution to PC2 (Figure 1B). All identified correlations are correlations of average strength.

Since there was no complete separation of the study groups, the group of smokers was divided into subgroups in accordance with the smoking experience (Figure 1C,D). Indicators characterizing the electrolyte balance (potassium, phosphorus, Ca/P ratio, Na/K ratio), AST/ALT ratio and superoxide dismutase (SOD) activity were selected as informative ones (Figure 1D). In this case, PC1 separates the groups Smokers I+Smokers II and Smokers III (Figure 1C). Phosphorus, potassium and AST/ALT ratio make a larger contribution to the separation (Figure 1E and Appendix A).

If we add a group of non-smokers to the calculations, then the centers of gravity of the ellipses corresponding to groups Smokers I+Smokers II coincide (Figure 1E). In this case, chlorides and catalase are added to the informative indicators (Figure 1F). The PC1 divides among themselves the groups of smokers and Non-smokers; phosphorus and chlorides make the greatest contribution to the separation (Appendix A). The greatest differences were found between the Non-smokers and Smokers III groups. The PC2 divides the groups by smoking experience, and the activity of antioxidant enzymes (catalase, SOD) and the AST/ALT ratio make the greatest contribution to the division (Figure 1F).

Thus, the biochemical indicators of saliva, which significantly change with smoking, include electrolyte components (phosphorus, potassium and chlorides), antioxidant enzymes and AST/ALT ratio. Taking into account the smoking experience makes it possible to switch from a generalized indicator (AOA) to specific enzymes: catalase and SOD.

### 2.3. The Salivary Biochemical Composition in Lung Cancer Patients with COPD

The next stage of the study was to identify the differences between lung cancer patients without COPD and with COPD of varying severity (Figure 2A,B). It was shown that, according to the Kruskal–Wallis criterion, the differences between the three groups are significant in pH (*p* = 0.0037), calcium content (*p* = 0.0117), sialic acids (*p* = 0.0459), LDH activity (*p* = 0.0100) and the level of diene conjugates (*p* = 0.0465). For data analysis by PCA, indicators with *p*-values ˂ 0.1000 were additionally selected according to the Kruskal–Wallis criterion (magnesium, catalase, Schiff bases, AST/ALT ratio). It was shown that PC1 divides the groups according to the presence/absence of COPD (*p* = 0.001, Figure 2A), with the greatest contribution made by LDH and catalase (Appendix A). PC2 divides the groups according to the severity of COPD (Figure 2B), and the most significant contribution to the separation is made by catalase and LDH, as well as sialic acids and Schiff bases (Figure 2B). The listed indicators of the salivary biochemical composition make it possible to differentiate between all three groups; however, the NO COPD and COPD II groups are the most different (Figure 2).

### 2.4. Features of the Salivary Biochemical Composition in Lung Cancer Patients with COPD Depending on Smoking

Each of the three groups (NO COPD, COPD I, COPD II) was divided according to the “yes/no” smoking status (Table 2). The table shows the values of all biochemical parameters that were significant for the separation of groups in at least one case. The parameters significant in accordance with the Kruskal–Wallis criterion were used further for the analysis by the PCA method (Figure 3). It was shown that PC1 divides the groups according to the presence/absence of COPD; the groups without COPD are located on the factor plane to the right of the vertical axis (Figure 3A). PC2 allows one to divide groups according to the severity of COPD; above the horizontal axis, there are groups with COPD I, below, with COPD II. In the absence of COPD, the horizontal axis differentiates the groups of smokers and non-smokers. Smokers are located on the factorial plane closer to the COPD II group, Non-smokers to COPD I. Chlorides, sialic acids, GGT, LDH and pH make the greatest contribution to the separation in PC1. Correlations with catalase, LDH and sialic acids were established for PC2 (Appendix A).

Thus, when dividing into groups, the main factor is the presence/absence of COPD with the maximum influence of metabolic enzymes (LDH, GGT) and sialic acids. For groups without COPD, the influence of smoking is manifested; significant factors are, as already shown in Section 2.2, chlorides and catalase.

It is shown that the nature of changes in indicators that determine the differences between the studied groups is ambiguous. Thus, regardless of smoking status, COPD II is characterized by the lowest activity of LDH (−39.5% and −42.4%), AST/ALT ratio (−14.3% and −18.1%) and catalase (−5.1% and −9.2% versus NO COPD group (smokers and non-smokers, respectively)). In the same group, the pH values are maximum, and the GGT activity is minimum for smokers with COPD II (−15.6%) and maximum for non-smokers (+4.1%). The content of sialic acids is minimal for the group of patients with COPD I, regardless of smoking (−27.3% and −35.6% compared with the NO COPD group, smokers and non-smokers, respectively). The level of uric acid differs maximally in the groups with COPD I non-smokers (minimum) and COPD II smokers (maximum). However, against the background of smoking and COPD, the level of uric acid is higher than in non-smokers with the corresponding severity of COPD, while in the NO COPD group, the level of uric acid against the background of smoking is lower than in the group of non-smokers. Chloride levels are higher in all groups of smokers, regardless of the presence/absence of COPD.

### 2.5. Metabolic Features of Salivary Composition in Lung Cancer and COPD, Depending on Smoking Experience

Simultaneous accounting of the presence/absence of COPD in lung cancer and smoking history shows that the previously noted trend continues. The centers of gravity of the ellipses corresponding to the Smokers I and Smokers II groups practically coincide (Figure 4A) but differ from the Non-smokers and Smokers III groups. The maximum differences are observed between the Non-smokers and Smokers III groups, which is quite natural. Thus, PCA analysis showed that the greatest contribution to PC1 is made by phosphorus, uric and pyruvic acids, while diene conjugates and catalase make the greatest contribution to PC2 (Appendix A).

If the group of patients without COPD is additionally divided in accordance with the smoking history, then groups without COPD are located to the right of the vertical axis on the factor plane and to the left with COPD, regardless of the smoking history (Figure 4C). It is interesting to note that for patients with COPD, the centers of gravity of the Smokers I and Smokers II ellipses coincide; without COPD, the differences between the same groups are noticeable. The Smokers III groups differ the most, without COPD and with COPD. The division into groups according to the presence/absence of COPD is determined by phosphorus, GGT and saliva pH (Appendix A).

Since the Smokers II group differs little from the Smokers I group, at the next stage, the PCA analysis was carried out for six groups: Smokers I, Smokers III and Non-smokers, depending on the presence/absence of COPD (Figure 4F). It was shown that NO COPD groups still remain on the right side of the vertical axis, but now a group with COPD+Smokers I is added to them. Uric and pyruvic acids make the largest contribution to PC1 (Appendix A). PC2 additionally divides the groups by smoking experience. Above the horizontal axis are the Non-smokers and Smokers I groups, below the axis are the Smokers III groups (Figure 4F). The maximum contribution to the separation of these groups is made by phosphorus and diene conjugates (Appendix A).

Thus, the vertical axis still divides into groups according to the presence/absence of COPD, and the horizontal axis according to the smoking history. At the same time, the differences between the Non-smokers and Smokers III groups are most pronounced for the group with COPD (Figure 4E,F).

### 2.6. Metabolic Features of Saliva Taking into Account the Type of COPD and Smoking History

While taking into account the severity of COPD and smoking experience, the groups are divided primarily according to their belonging to the COPD I and COPD II groups (Figure 5A). Thus, all groups with COPD I, regardless of smoking history, are located to the right of the vertical axis, while groups with COPD II are located to the left. The horizontal axis divides the groups by smoking experience: the maximum differences were found for the groups Smokers I and Smokers III, while for Non-smokers, regardless of the type of COPD, intermediate values were established. For PC1, high correlation coefficients with pH, chlorides and calcium are shown, while for PC2 with diene conjugates and catalase (Appendix A). In general, it should be noted that for COPD I, the differences between subgroups with different smoking histories are less pronounced than for COPD II. A detailed change in all studied parameters of saliva in groups of patients with lung cancer in combination with COPD, depending on the smoking history, is shown in Table 3.

### 2.7. The Prognostic Value of Smoking in Groups of Patients with Lung Cancer and COPD of Varying Severity

Differences in overall survival depending on smoking were revealed only for groups without COPD (Figure 6a), while the median of OS for non-smokers was 20.7 months and for smokers 14.8 months (HR = 1.45, 95% CI 0.83–2.52, *p* = 0.07368). For patients with COPD, the differences between Smokers and Non-smokers are reflected only in 5-year survival (Figure 6c); the median of OS for Non-smokers was 16.8 months and for smokers 20.5 months (HR = 1.27, 95% CI 0.69–2.31, *p* = 0.54182).

For patients with lung cancer without COPD, there were pronounced differences between groups with different smoking histories (Figure 6b). Smokers I have a median overall survival of 28.2 months, Smokers II 14.9 months and Smokers III 12.8 months; the relative risk for Smokers II was HR_1–2_ = 1.75 (95% CI 0.55–5.49, *p* = 0.36863), while for the Smokers III HR_1–3_ = 4.67 (95% CI 1.01–21.13, *p* = 0.03838) (Figure 6d). OS for patients with lung cancer and COPD for Smokers I was 32.0 months, for Smokers II 16.9 months (HR_1–2_ = 3.73, 95% CI 1.31–10.39, *p* = 0.01956) and for Smokers III 15.1 months (HR_1–3_ = 4.43, 95% CI 1.40–16.38, *p* = 0.00956).

Taking into account the severity of COPD, the same trend persists: for Smokers I, the median OS was 32.0 and 35.6 months for COPD I and COPD II, respectively (Figure 6e,f). For Smokers II, the median OS sharply decreases to 17.7 and 16.9 months (HR_1–2_ = 3.42, 95% CI 0.89–12.96, *p* = 0.09723 and HR_1–2_ = 3.67, 95% CI 0.18–75.31, *p* = 0.27084). For Smokers III, the median OS was 14.2 and 16.0 months (HR_1–3_ = 3.67, 95% CI 1.18–11.22, *p* = 0.06377 and HR_1–3_ = 2.83, 95% CI 0.19–40.95, *p* = 0.36018) (Figure 6e,f).

## 3. Discussion

The study of the metabolic profile of biological fluids, including saliva, in lung cancer in combination with COPD and the simultaneous consideration of smoking history has not been conducted before. In this regard, it is difficult to compare the data obtained by us with the literature data.

It was shown that when the smoking factor and the presence of COPD in lung cancer are taken into account separately, the differences between the subgroups characterize different groups of saliva biochemical markers. Thus, the differences between smokers and non-smokers are due to the activity of antioxidant enzymes (catalase, SOD and AOA) and the content of electrolyte components of saliva (chlorides, potassium and phosphorus). For COPD, the maximum differences between subgroups are determined by the activity of metabolic enzymes (LDH, AST/ALT ratio), the level of lipid peroxidation products (diene conjugates and Schiff bases), the content of sialic acids and electrolyte components (pH, calcium and magnesium). When dividing a group of patients with COPD into subgroups in accordance with the severity of COPD, the differences between the groups remain primarily in terms of the presence/absence of COPD. The centers of gravity of the ellipses for smokers and non-smokers with the same severity of COPD are located close to each other on the factor diagram, which in this case demonstrates the secondary influence of the smoking factor on the biochemical parameters of saliva. A number of studies have shown that, depending on the presence/absence of COPD, the molecular and morphological features of lung cancer differ [30]. Taking into account additional smoking history, we again see that the first main component classifies lung cancer patients according to the presence/absence of COPD regardless of smoking history. However, in this case, additional informative parameters appear: uric and pyruvic acids. For these indicators, there is a different direction of change in the groups with and without COPD. Thus, the level of uric acid is higher in the groups of smokers with COPD but lower in the group of smokers without COPD. The opposite is true for pyruvic acid. Taking into account COPD and smoking experience, paradoxical data were obtained: the main component divided all patients into COPD I and COPD II groups, and the subsequent division showed that the groups of nonsmoking patients occupy an intermediate position between Smokers I and Smokers III in terms of biochemical parameters of saliva.

The literature confirms that lung cancer among non-smokers exhibits distinctive clinical characteristics, is more common in women and is diagnosed at later stages and that adenocarcinoma is the predominant histological type [31,32,33]. Meanwhile, tobacco smoking is associated with squamous and small cell types of lung cancer, as well as with an earlier age at the time of diagnosis [34], which leads to a generally poorer prognosis for patients in this group. The revealed differences between the predominant histological types of lung cancer and the gender composition of smokers and non-smokers are apparently due to a decrease in the prevalence of smoking, as well as a decrease in the content of tar and carcinogenic substances in tobacco smoke [35]. It is known that the gap between tobacco use and the development of lung cancer can be up to 30–40 years [36]. Since smoking is one of the factors in the development of COPD, more severe stages of COPD should be detected in patients with long smoking history. We showed this when studying the structure of the studied group (Table 1).

A causal relationship between COPD and lung carcinogenesis is not yet known. It is generally accepted that chronic inflammation plays a central role in the pathogenesis of COPD and lung tumorigenesis. Cigarette smoke is known to affect processes associated with inflammation such as angiogenesis, autophagy/apoptosis and chromatin remodeling, which are critical for the development of COPD and cancer. The fact that the inflammation accompanying COPD is reflected in the composition of saliva is confirmed by studies that show that a feature of the cellular composition of salivary immunocytes in smokers with early forms of COPD is the prevalence of the helper population and, accordingly, a high ratio of CD4+/CD8+, and in this group, more high content of IL-17 in saliva in comparison with the comparison group—smokers without signs of COPD [37]. In our study, we see a change in the activity of metabolic enzymes, in particular, a decrease in the activity of LDH, GGT and aminotransferases during COPD progression, as well as a decrease in the activity of antioxidant enzymes (catalase, AOA). We have previously shown that a decrease in the activity of LDH in saliva is a prognostically unfavorable sign in lung cancer [17]. According to the literature, 35 altered and common metabolites were identified between patients with lung cancer and COPD, including amino acids, fatty acids, lysophospholipids, phospholipids and triacylglycerides, with the metabolism of alanine, aspartate and glutamate being the most altered [38], which is consistent with our data.

An independent comparison of overall survival rates for the same groups of patients showed that for Smokers I, regardless of the presence/absence and type of COPD, the overall survival rates are significantly better (Figure 6). The very fact of having COPD results in no difference in overall survival rates between Smokers and Non-smokers with lung cancer.

The revealed metabolic changes in saliva, in particular, an increase in the activity of antioxidant enzymes, can be used to monitor patients from risk groups for early diagnosis of lung cancer. It is also potentially possible to adjust the treatment process, in particular, therapy aimed at reducing the level of antioxidants [39,40]. However, these areas require serious study.

The limitations of the study include the absence of groups with moderate and severe COPD, as well as a small number of patients in some groups, which does not allow comparison of the studied parameters in accordance with the histological type of lung cancer or the stage of the disease.

## 4. Materials and Methods

### 4.1. Study Design and Group Description

We included 392 patients with lung cancer who were hospitalized in the thoracic department of the Clinical Oncological Dispensary in Omsk in the period 2014–2017. The inclusion criteria were: the age of the patients 30–75 years, the absence of any treatment at the time of inclusion in the study, including surgery, chemotherapy or radiation, histological verification of the diagnosis. The collection of saliva samples was carried out strictly before the start of treatment. Lung cancer of various histological types was confirmed in all patients, including: adenocarcinoma (ADC, n = 189), squamous cell carcinoma (SCC, n = 135) and neuroendocrine cancer (NEC, n = 68) (Table 4).

To describe the severity of COPD, a classification based on forced expiratory volume in the first second (FEV1) as a percentage of predicted (FEV1% pred) was used [41]. GOLD criteria were used to classify the severity of COPD (GOLD I; FEV1 > 80% of predicted, GOLD II; FEV1 = 50–79% of predicted, GOLD III; FEV1 = 30–49% of predicted, and GOLD IV; FEV1 < 30% of predicted). In accordance with the above classification, 114 patients were assigned to the GOLD I type, 51 patients were assigned to the GOLD II type and 4 patients were assigned to the GOLD III type (Table 1). In 223 patients, no COPD was detected. In this study, we examined groups of patients without COPD, with mild and moderate COPD (GOLD I, GOLD II), designated, respectively, NO COPD, COPD I and COPD II. Additional division of these into subgroups in accordance with the histological type of lung cancer was not carried out since we have previously shown that the factor of the presence/absence of COPD is more significant for changes in the metabolic profile of saliva than the histological type of lung cancer [29]. All patients with COPD were out of the exacerbation stage, and therefore non-drug support measures were implemented and short-acting bronchodilators were prescribed for use as needed. None of the patients were receiving corticosteroid treatment for COPD at the time of study entry.

Participants also completed a questionnaire regarding tobacco smoking. To calculate the smoking history, we used the following data: number of cigarettes smoked per day divided by 20 multiplied by the total number of years of smoking (results in packs/year). For each of the selected groups, a division into subgroups was carried out in accordance with the presence/absence of smoking and its experience: Non-smokers, Smokers I (less than 24 pack-years), Smokers II (25–49 pack-years), Smokers III (more than 50 pack-years). The group of non-smokers included patients who had never smoked, as well as those who stopped smoking more than ten years ago.

The study was carried out in accordance with the Helsinki Declaration (adopted in June 1964 in Helsinki, Finland, and revised in October 2000 in Edinburgh, Scotland) and was approved at a meeting of the Ethics Committee of the Omsk Regional Clinical Hospital “Clinical Oncology Center” on 21 July 2016 (Protocol No. 15). All of the volunteers provided written informed consent.

### 4.2. Collection, Processing and Storage of Saliva Samples

Saliva (5 mL) was collected from all participants prior to treatment. Collection of saliva samples was carried out on an empty stomach after rinsing the mouth with water at 8–10 a.m. by spitting into sterile polypropylene tubes; the salivation rate (mL/min) was calculated. Patients who smoked abstained from smoking for 1–3 h before collecting saliva [42]. Saliva samples were centrifuged (10,000× *g* for 10 min) (CLb-16, Moscow, Russia), after which biochemical analysis was immediately performed without storage and freezing.

### 4.3. Biochemical Analysis of Saliva

The biochemical composition of the samples was established using the StatFax 3300 semi-automatic biochemical analyzer (Awareness Technology, Palm City, FL, USA) [17]. The pH, mineral composition (calcium, phosphorus, sodium, potassium, magnesium, chlorides), the content of urea, total protein, albumin, uric acid, α-amino acids, imidazole compounds, seromucoids and sialic acids and the activity of enzymes (aminotransferases (ALT, AST); alkaline phosphatase (ALP); lactate dehydrogenase (LDH); gamma-glutamyl transpeptidase (GGT); α-amylase) were determined in all samples. In all samples, the content of substrates for lipid peroxidation processes (diene conjugates, triene conjugates, Schiff bases, malondialdehyde MDA) was determined. Additionally, we assessed the activity of antioxidant enzymes (catalase, superoxide dismutase, antioxidant activity).

### 4.4. Statistical Analysis

Statistical analysis was performed using Statistica 13.3 EN software (StatSoft, Tulsa, OK, USA); R version 3.6.3; RStudio Version 1.2.5033; FactoMineR version 2.3. (RStudio, version 3.2.3, Boston, MA, USA) by a nonparametric method using the Mann–Whitney U-test and the Kruskal–Wallis H-test. The description of the sample was made by calculating the median (Me) and interquartile range in the form of the 25th and 75th percentiles [LQ; UQ]. Differences were considered statistically significant at *p* ˂ 0.05.

A principal component analysis (PCA) was performed using the PCA program in R [43]. The choice of variables for the PCA method was carried out according to the results of comparison of biochemical indicators in the studied groups. When comparing two groups, we used the Mann–Whitney test; when comparing three groups or more, we used the Kruskal–Wallis test. Next, we selected indicators for which the differences between all groups are significant at the *p* ˂ 0.10 level. PCA results are presented in the form of factor planes and corresponding correlation circles. In each case, the figures show only the first two principal components (PC1 and PC2). The color of the arrows on the correlation circle changes from blue (weak correlation) to red (strong correlation) as shown on the color bar. The orientation of the arrows characterizes positive and negative correlations (for the first principal component, we analyze the location of the arrows relative to the vertical axis; for the second principal component, relative to the horizontal axis). A complete list of the values of the correlation coefficients for the principal components is given in the Appendix A. The significance of the correlation is determined by the correlation coefficient (r): strong—r = ±0.700 to ±1.00, medium—r = ±0.300 to ±0.699, weak—r = 0.00 to ±0.299.

The survival curve was calculated by the Kaplan–Meier method and compared using the Log-rank test for univariate analysis (Statistica 10.0, StatSoft). Prognostic factors were analyzed by multivariate analysis using Cox’s proportional hazard model in a backward stepwise fashion to adjust for potential confounding factors. Overall survival (OS) was computed from the date of diagnosis to the date of death or the date of last follow-up. Survival data were obtained until December 2019.

## 5. Conclusions

Saliva is a suitable biological fluid for studying metabolic processes in lung cancer. We showed metabolic features of lung cancer in combination with COPD of varying severity and smoking for the first time. When the factor of smoking and the presence of COPD in lung cancer are taken into account separately, different groups of biochemical indicators of saliva are informative. It was shown that, depending on the smoking status, the activity of the antioxidant enzymes of saliva changes primarily, while in COPD the changes concern the activity of metabolic enzymes, which confirms the deeper nature of the changes in COPD. With the simultaneous consideration of the factor of smoking and the presence of COPD, biochemical indicators corresponding to the presence/absence and severity of COPD are a priority. Changes occurring against the background of smoking are of a secondary nature, being manifested as much as possible with a smoking experience of more than 50 pack-years. In the presence of COPD as a concomitant pathology, smoking cessation does not affect the overall survival rate for lung cancer.

## Figures and Tables

**Figure 1 metabolites-11-00289-f001:**
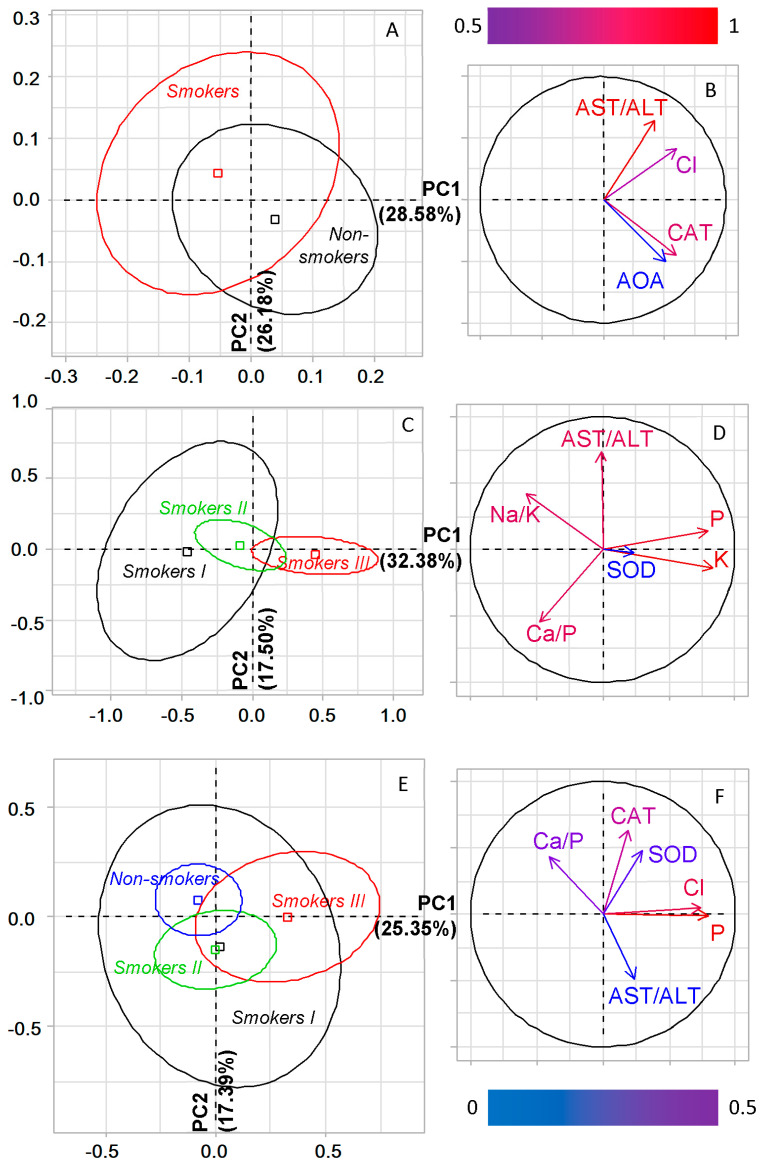
PCA. Factor plane (**A**,**C**,**E**) and correlation circle (**B**,**D**,**F**) for groups of lung cancer patients without taking into account COPD: (**A**,**B**)—smoking yes/no; (**C**,**D**)—only smokers, taking into account smoking experience; (**E**,**F**)—smokers and non-smokers, taking into account the smoking experience. Here and in Figure 2, Figure 3, Figure 4 and Figure 5, the color of the arrows on the correlation circle changes from blue (weak correlation) to red (strong correlation) as shown on the color bar. The orientation of the arrows characterizes positive and negative correlations (for the first principal component, we analyze the location of the arrows relative to the vertical axis; for the second principal component, relative to the horizontal axis). CAT—catalase, AOA—antioxidant activity, Cl—chlorides, AST/ALT—aspartate aminotransferase to alanine aminotransferase ratio, SOD—superoxide dismutase, P—phosphorus, K—potassium, Na/K—sodium to potassium ratio, Ca/P—calcium to phosphorus ratio.

**Figure 2 metabolites-11-00289-f002:**
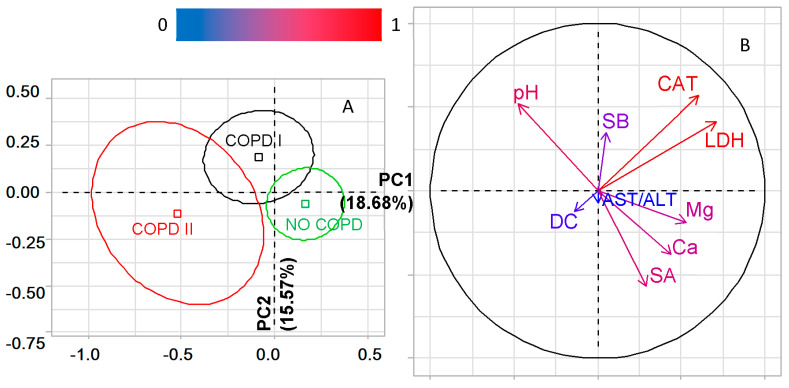
PCA. Factor plane (**A**) and correlation circle (**B**) for patient groups, taking into account the presence/absence and severity of COPD. CAT—catalase, LDH—lactate dehydrogenase, AST/ALT—aspartate aminotransferase to alanine aminotransferase ratio, SB—Schiff bases, DC—diene conjugates, SA—sialic acids, Ca—calcium, Mg—magnesium.

**Figure 3 metabolites-11-00289-f003:**
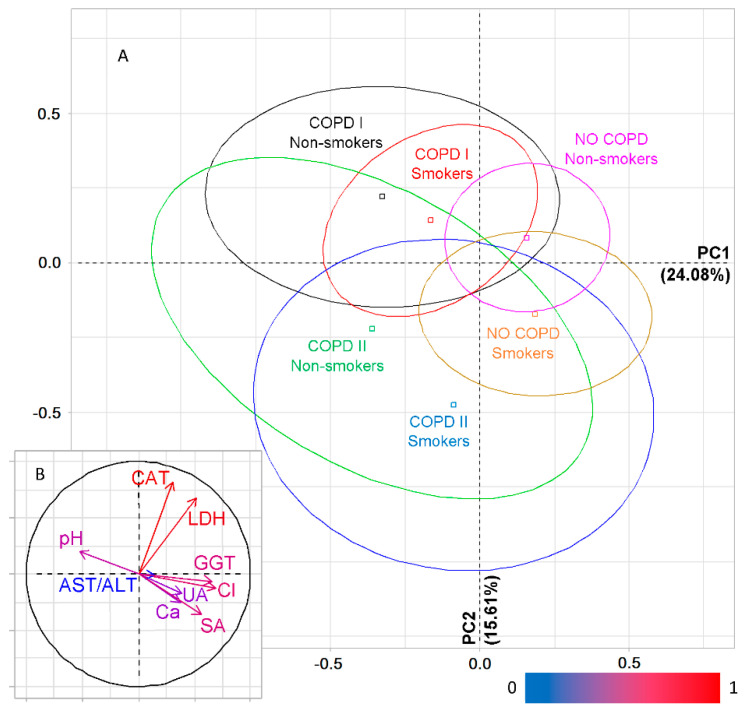
PCA. Factor plane (**A**) and correlation circle (**B**) for dividing groups of smokers and non-smokers in patients with lung cancer and COPD of varying severity. CAT—catalase, LDH—lactate dehydrogenase, GGT—gamma glutamyltransferase, AST/ALT—aspartate aminotransferase to alanine aminotransferase ratio, SA—sialic acids, UA—uric acid, Cl—chlorides, Ca—calcium.

**Figure 4 metabolites-11-00289-f004:**
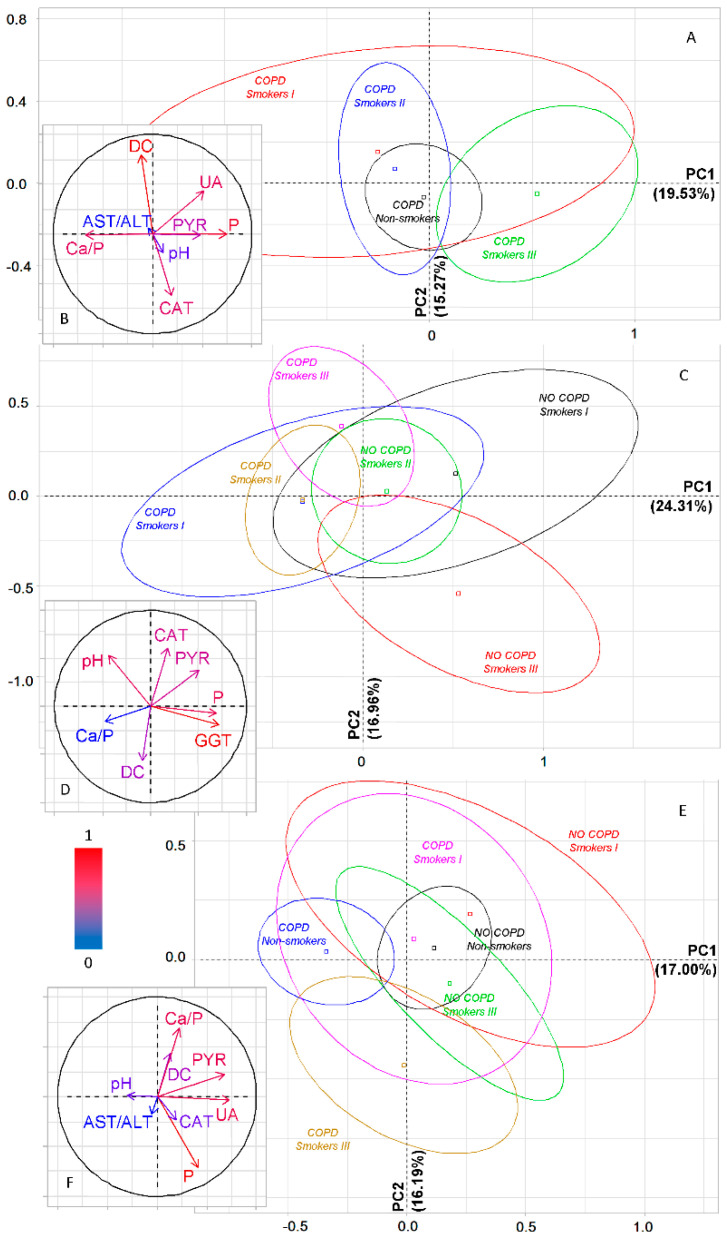
PCA. Factor plane (**A**,**C**,**E**) and correlation circle (**B**,**D**,**F**) when taking into account the experience of smoking: (**A**,**B**)—only COPD, taking into account the experience of smoking (Non-smokers, Smokers I, Smokers II and Smokers III); (**C**,**D**)—6 groups with/without COPD, taking into account smoking experience (COPD Yes/No+Smokers I, Smokers II and Smokers III); (**E**,**F**)—6 groups including non-smokers without group Smokers II (COPD I/II+Non-smokers, Smokers I and Smokers III). CAT—catalase, AST/ALT—aspartate aminotransferase to alanine aminotransferase ratio, GGT—gamma glutamyltransferase, P—phosphorus, Ca/P—calcium to phosphorus ratio, UA—uric acid, PYR—pyruvic acid, DC—diene conjugates.

**Figure 5 metabolites-11-00289-f005:**
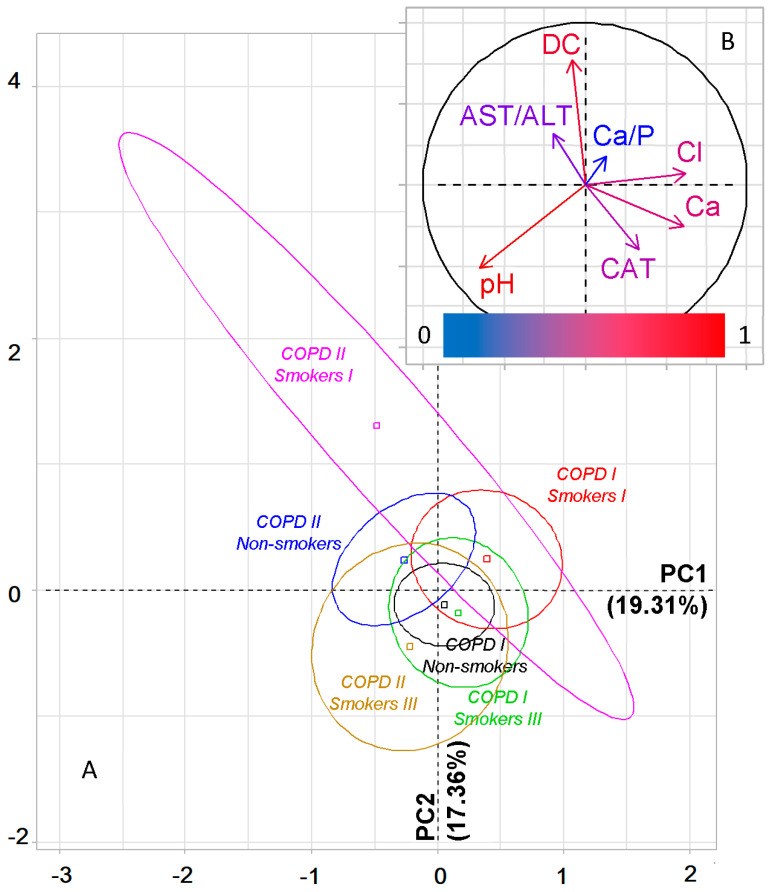
PCA. Factor diagram (**A**) and correlation circle (**B**) for dividing patient groups depending on smoking history and severity of COPD. CAT—catalase, AST/ALT—aspartate aminotransferase to alanine aminotransferase ratio, Cl—chlorides, Ca—calcium, Ca/P—calcium to phosphorus ratio, DC—diene conjugates.

**Figure 6 metabolites-11-00289-f006:**
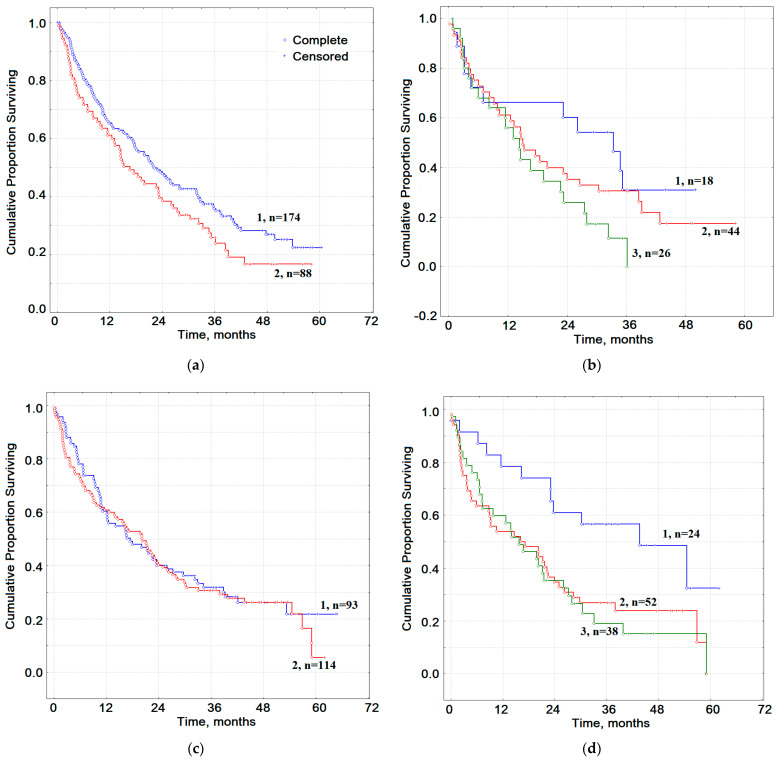
(**a**) OS for patients with LC+NO COPD: curve 1—Non-smokers, curve 2—Smokers; (**b**) OS for patients with LC+NO COPD: curve 1—Smokers I, curve 2—Smokers II, curve 3—Smokers III; (**c**) OS for patients with LC+COPD: curve 1—Non-smokers, curve 2—Smokers; (**d**) OS for patients with LC+COPD: curve 1—Smokers I, curve 2—Smokers II, curve 3—Smokers III; (**e**) OS for patients with LC+COPD I: curve 1—Smokers I, curve 2—Smokers II, curve 3—Smokers III; (**f**) OS for patients with LC+COPD II: curve 1—Smokers I, curve 2—Smokers II, curve 3—Smokers III.

**Table 1 metabolites-11-00289-t001:** The structure of the study groups depending on smoking and type of COPD.

Smoking Status	Subgroups	Number of Patients		
		NO COPD	COPD I	COPD II
**Non-smokers**	n (%)	147 (65.9)	49 (43.0)	23 (45.1)
	Age, years	60.0 [55.0; 64.5]	57.0 [53.0; 61.0]	63.0 [58.0; 65.0]
**Smokers I**	n (%)	14 (6.3)	16 (14.0)	4 (7.8)
	Age, years	57,5 [48.0; 60.0]	54.0 [52.0; 59.0]	59.5 [54.5; 65.5]
**Smokers II**	n (%)	39 (17.5)	35 (30.7)	10 (19.6)
	Age, years	62.0 [55.0; 63.0]	59.0 [54.0; 64.0]	60.5 [55.5; 63.5]
**Smokers III**	n (%)	23 (10.3)	14 (12.3)	14 (27.5)
	Age, years	63.5 [61.0; 68.0] *^,^**	62.0 [59.0; 66.0] **	65.0 [61.0; 69.0] **

Note. * The differences between the groups Smokers II and Smokers III are statistically significant (*p* = 0.0140). ** The differences between the groups Smokers I and Smokers III are statistically significant (NO COPD: *p* = 0.0015, COPD I: *p* = 0.0026, COPD II: *p* = 0.0125). Age is presented as median and interquartile range: Me [LQ; UQ].

**Table 2 metabolites-11-00289-t002:** Biochemical markers of saliva in lung cancer patients with COPD of varying severity depending on smoking status.

Indicators	Smoking	LC+SM, n = 76	COPDI+SM, n = 65	COPDII+SM, n = 28	Kruskal–Wallis Test (H, *p*)
		LC+NO SM, n = 147	COPDI+NO SM, n = 49	COPDII+NO SM, n = 23	
**Electrolytes**					
pH	YES	6.44 [6.22; 6.64]	6.60 [6.35; 6.82]	6.60 [6.29; 6.91]	11.55, 0.0415 *
	NO	6.47 [6.20; 6.71]	6.52 [6.22; 6.88]	6.68 [6.21; 7.06]	
Calcium, mmol/L	YES	1.50 [1.03; 2.13]	1.28 [0.86; 1.80]	1.35 [0.99; 1.76]	11.17, 0.0481 *
	NO	1.44 [1.07; 1.80]	1.48 [1.03; 1.82]	1.21 [0.81; 1.47]	
Phosphorus, mmol/L	YES	4.50 [3.28; 6.26]	4.07 [3.10; 5.43]	4.90 [3.51; 6.69]	5.405, 0.3685
	NO	4.62 [3.53; 5.72]	4.61 [3.25; 5.54]	4.26 [3.17; 4.98]	
Ca/P ratio, c.u.	YES	0.32 [0.22; 0.48]	0.30 [0.18; 0.50]	0.28 [0.17; 0.38]	4.765, 0.4452
	NO	0.32 [0.23; 0.47]	0.35 [0.23; 0.48]	0.28 [0.20; 0.44]	
Potassium, mmol/L	YES	13.2 [8.9; 16.3]	13.1 [9.9; 14.9]	14.3 [10.8; 19.9]	4.469, 0.4841
	NO	12.5 [9.9; 15.7]	12.4 [8.3; 16.9]	12.5 [8.5; 15.8]	
Na/K ratio, c.u.	YES	0.72 [0.45; 1.50]	0.89 [0.48; 1.52]	0.57 [0.48; 1.01]	3.871, 0.5682
	NO	0.75 [0.47; 1.13]	0.70 [0.44; 0.94]	0.84 [0.46; 1.39]	
Chlorides, mmol/L	YES	28.9 [22.3; 37.3]	29.5 [24.0; 38.0]	31.6 [21.0; 36.9]	9.752, 0.0826 **
	NO	28.0 [22.3; 33.7]	24.7 [18.0; 31.0]	28.0 [20.0; 34.6]	
Magnesium, mmol/L	YES	0.311 [0.231; 0.383]	0.291 [0.228; 0.339]	0.302 [0.200; 0.372]	6.395, 0.2697
	NO	0.311 [0.246; 0.390]	0.286 [0.219; 0.379]	0.277 [0.231; 0.301]	
**Protein Metabolism**					
Uric acid, μmol/L	YES	67.4 [35.3; 166.4]	79.1 [37.1; 159.1]	113.9 [34.5; 196.5]	10.83, 0.0550 **
	NO	97.6 [46.3; 180.8]	53.5 [22.3; 121.3]	94.0 [34.6; 180.8]	
Sialic acids, mmol/L	YES	0.201 [0.119; 0.287]	0.146 [0.085; 0.229]	0.183 [0.095; 0.317]	9.977, 0.0759 **
	NO	0.177 [0.104; 0.287]	0.114 [0.061; 0.269]	0.189 [0.092; 0.317]	
Pyruvic acid, μmol/L	YES	16.91 [10.54; 25.00]	13.48 [9.31; 18.63]	12.62 [8.82; 16.91]	6.512, 0.2595
	NO	15.20 [10.29; 21.81]	14.22 [8.33; 18.87]	13.97 [8.82; 16.67]	
**Enzymes**					
AST/ALT ratio, c.u.	YES	1.19 [0.88; 1.54]	1.29 [1.02; 1.75]	1.02 [0.76; 1.56]	10.02, 0.0746 **
	NO	1.38 [1.04; 1.73]	1.33 [0.97; 1.66]	1.13 [1.01; 1.34]	
LDH, U/L	YES	1144.0 [615.9; 1619.0]	1248.5 [625.3; 1715.5]	691.6 [462.3; 1254.1]	9.959, 0.0764 **
	NO	1265.0 [524.2; 2022.0]	1110.0 [605.2; 1761.7]	728.6 [469.3; 1277.5]	
GGT, U/L	YES	23.7 [19.4; 26.8]	21.6 [17.8; 25.1]	20.0 [18.1; 22.6]	9.427, 0.0932 **
	NO	22.0 [18.2; 26.0]	20.2 [16.0; 26.0]	22.9 [19.0; 25.4]	
Catalase, ncat/mL	YES	2.53 [1.98; 3.92]	2.69 [2.02; 3.85]	2.40 [1.80; 2.95]	10.16, 0.0708 **
	NO	2.82 [2.14; 4.56]	2.80 [2.32; 4.17]	2.56 [1.72; 2.82]	
SOD, c.u.	YES	52.6 [18.4; 100.0]	63.2 [26.3; 110.5]	73.7 [42.1; 86.8]	3.927, 0.5600
	NO	68.4 [28.9; 136.8]	68.4 [50.0; 113.2]	63.2 [28.9; 139.5]	
AOA, mmol/L	YES	1.92 [1.38; 2.14]	1.83 [1.41; 2.14]	1.70 [1.60; 1.86]	5.722, 0.3342
	NO	1.67 [1.49; 1.85]	1.78 [1.49; 1.87]	1.40 [1.14; 1.90]	
**Lipoperoxidation Products**					
DC, c.u.	YES	4.06 [3.88; 4.22]	3.92 [3.78; 4.18]	3.92 [3.75; 4.09]	8.655, 0.1236
	NO	3.99 [3.78; 4.16]	3.93 [3.68; 4.11]	3.99 [3.85; 4.23]	
SB, c.u.	YES	0.564 [0.501; 0.670]	0.538 [0.480; 0.656]	0.574 [0.501; 0.661]	5.069, 0.4075
	NO	0.561 [0.488; 0.676]	0.531 [0.463; 0.654]	0.571 [0.534; 0.665]	
MDA, μmol/L	YES	7.01 [5.90; 9.32]	6.32 [5.56; 8.89]	8.12 [6.07; 10.85]	8.473, 0.1320
	NO	7.09 [5.73; 9.06]	6.92 [5.64; 9.40]	8.50 [5.90; 10.13]	

Note. LC+SM—lung cancer+smokers without COPD; LC+NO SM—lung cancer+non-smokers without COPD; COPDI+SM—lung cancer+COPD I+smokers; COPDI+NO SM—lung cancer+COPD I+non-smokers; COPDII+SM—lung cancer+COPD II+smokers; COPDII+NO SM—lung cancer+COPD II+non-smokers. * Differences between 6 groups are statistically significant, *p* ˂ 0.05. ** Differences between 6 groups are statistically significant, *p* ˂ 0.10. Na/K—sodium to potassium ratio, AST/ALT—aspartate aminotransferase to alanine aminotransferase ratio, LDH—lactatedehydrogenase, GGT—gamma glutamyltransferase, SOD—superoxide dismutase, AOA—antioxidant activity, DC—diene conjugates, SB—Schiff bases, MDA—malondialdehyde.

**Table 3 metabolites-11-00289-t003:** The biochemical composition of the saliva of patients with lung cancer, depending on the smoking history and the severity of COPD.

Indicator	COPD Type	Smokers I n = 14/16/4	Smokers II n = 39/35/10	Smokers III n = 23/14/14	Kruskal–Wallis Test (H, *p*)
**Electrolytes**					
pH	NO COPD	6.41 [6.18; 6.55]	6.48 [6.24; 6.69]	6.36 [6.02; 6.64]	15.35, 0.1670
	COPD I	6.66 [6.32; 6.78]	6.56 [6.38; 6.82]	6.61 [6.35; 7.07]	
	COPD II	6.54 [6.19; 6.75]	6.57 [6.23; 6.80]	6.62 [6.35; 7.17]	
Calcium, mmol/L	NO COPD	1.49 [0.91; 2.24]	1.56 [1.17; 2.17]	1.28 [0.97; 2.08]	15.75, 0.1507
	COPD I	1.53 [0.85; 1.90]	1.26 [0.75; 1.79]	1.28 [1.11; 1.68]	
	COPD II	1.31 [0.68; 2.24]	1.46 [1.10; 1.99]	1.21 [0.90; 1.52]	
Phosphorus, mmol/L	NO COPD	4.32 [2.42; 4.79]	4.68 [3.60; 6.04]	5.01 [2.94; 8.84]	13.75, 0.2473
	COPD I	4.35 [3.16; 5.56]	3.82 [2.93; 5.09]	4.79 [3.58; 5.91]	
	COPD II	4.70 [1.76; 7.37]	3.92 [3.29; 5.67]	5.68 [4.33; 7.76]	
Ca/P ratio, c.u.	NO COPD	0.35 [0.30; 0.49]	0.33 [0.26; 0.51]	0.24 [0.15; 0.42]	15.39, 0.1653
	COPD I	0.35 [0.16; 0.54]	0.29 [0.21; 0.51]	0.30 [0.18; 0.41]	
	COPD II	0.30 [0.20; 6.57]	0.34 [0.28; 0.47]	0.24 [0.13; 0.30]	
Potassium, mmol/L	NO COPD	8.9 [6.7; 15.0]	13.1 [9.2; 15.9]	13.5 [8.9; 19.2]	9.458, 0.5797
	COPD I	12.5 [7.3; 14.1]	14.1 [10.6; 15.9]	13.0 [8.4; 14.9]	
	COPD II	12.1 [8.5; 13.8]	12.7 [10.3; 14.7]	17.5 [12.3; 21.1]	
Na/K ratio, c.u.	NO COPD	1.70 [0.45; 2.00]	0.67 [0.40; 1.50]	0.72 [0.50; 1.33]	12.47, 0.3293
	COPD I	1.19 [0.64; 1.55]	0.60 [0.39; 1.30]	1.08 [0.59; 2.29]	
	COPD II	1.01 [0.65; 1.29]	0.65 [0.50; 1.03]	0.49 [0.36; 0.82]	
Chlorides, mmol/L	NO COPD	28.9 [22.9; 41.7]	29.2 [22.8; 37.6]	25.1 [21.2; 35.6]	15.65, 0.1546
	COPD I	34.7 [26.2; 38.8]	29.1 [22.5; 37.0]	29.5 [22.9; 38.4]	
	COPD II	23.1 [16.8; 29.1]	35.6 [31.6; 38.9]	28.5 [20.4; 35.1]	
Magnesium, mmol/L	NO COPD	0.317 [0.188; 0.368]	0.311 [0.257; 0.384]	0.280 [0.221; 0.381]	7.678, 0.7419
	COPD I	0.317 [0.248; 0.339]	0.294 [0.232; 0.358]	0.281 [0.197; 0.336]	
	COPD II	0.256 [0.201; 0.335]	0.309 [0.228; 0.442]	0.305 [0.164; 0.372]	
**Protein Metabolism**					
Uric acid, μmol/L	NO COPD	84.4 [40.5; 167.2]	63.1 [28.3; 165.5]	61.9 [36.3; 160.9]	14.59, 0.2019
	COPD I	94.5 [29.7; 176.3]	69.5 [41.0; 130.0]	93.5 [34.7; 211.5]	
	COPD II	49.3 [29.5; 125.9]	109.4 [56.6; 172.4]	144.4 [40.8; 283.9]	
Sialic acids, mmol/L	NO COPD	0.171 [0.098; 0.201]	0.207 [0.137; 0.281]	0.183 [0.119; 0.317]	12.35, 0.3381
	COPD I	0.146 [0.101; 0.204]	0.116 [0.067; 0.232]	0.183 [0.104; 0.226]	
	COPD II	0.122 [0.098; 0.317]	0.177 [0.078; 0.397]	0.207 [0.095; 0.323]	
Pyruvic acid, μmol/L	NO COPD	17.65 [11.27; 32.60]	13.11 [8.82; 26.23]	19.24 [14.46; 22.79]	15.20, 0.1735
	COPD I	14.83 [8.58; 27.08]	11.03 [8.70; 15.81]	16.67 [13.48; 22.79]	
	COPD II	12.75 [7.23; 48.16]	14.22 [10.05; 17.40]	11.64 [7.84; 16.42]	
**Enzymes**					
AST/ALT ratio, c.u.	NO COPD	1.25 [1.04; 1.61]	1.19 [0.87; 1.57]	1.08 [0.77; 1.38]	15.28, 0.1699
	COPD I	1.33 [1.07; 1.77]	1.35 [1.03; 1.75]	1.10 [0.82; 1.35]	
	COPD II	1.07 [0.84; 14.30]	1.28 [0.75; 1.59]	0.97 [0.76; 1.50]	
LDH, U/L	NO COPD	1177.0 [535.9; 1958.5]	1251.0 [649.7; 1549.0]	907.6 [604.3; 1575.0]	12.65, 0.3170
	COPD I	1494.0 [697.8; 2321.0]	981.0 [492.4; 1586.5]	1278.5 [882.0; 1610.0]	
	COPD II	547.7 [223.0; 1006.4]	729.3 [488.5; 1197.0]	726.3 [484.1; 1496.0]	
GGT, U/L	NO COPD	25.0 [20.7; 31.2]	22.3 [19.5; 26.0]	24.4 [18.8; 27.6]	12.85, 0.3033
	COPD I	20.2 [18.0; 24.5]	20.0 [17.4; 25.2]	22.6 [21.1; 23.3]	
	COPD II	18.8 [16.7; 24.2]	21.5 [19.3; 22.6]	19.7 [17.9; 22.2]	
Catalase, ncat/mL	NO COPD	3.48 [2.52; 5.29]	2.22 [2.00; 3.86]	2.44 [1.53; 2.97]	20.10, 0.0440 *
	COPD I	3.00 [1.76; 4.28]	2.60 [2.10; 3.85]	2.70 [2.20; 4.12]	
	COPD II	2.16 [1.75; 2.81]	2.12 [1.39; 2.64]	2.61 [1.98; 3.18]	
SOD, c.u.	NO COPD	36.8 [13.2; 68.4]	57.9 [18.4; 100.0]	61.8 [27.6; 111.8]	11.86, 0.3742
	COPD I	50.0 [26.3; 71.1]	65.8 [34.2; 160.5]	57.9 [18.4; 165.8]	
	COPD II	43.4 [39.5; 47.4]	52.6 [40.8; 85.5]	84.2 [67.1; 160.5]	
AOA, mmol/L	NO COPD	1.84 [1.32; 2.26]	1.78 [1.45; 2.07]	2.09 [1.92; 2.23]	10.22, 0.5110
	COPD I	2.03 [1.70; 2.10]	1.67 [1.38; 2.10]	2.13 [1.61; 2.99]	
	COPD II	No data	1.76 [1.52; 1.99]	1.70 [1.65; 1.78]	
**Lipoperoxidation Products**					
DC, c.u.	NO COPD	4.08 [3.82; 4.27]	3.99 [3.86; 4.16]	4.09 [3.98; 4.25]	20.94, 0.0340 *
	COPD I	3.94 [3.81; 4.12]	4.01 [3.76; 4.21]	3.86 [3.70; 4.02]	
	COPD II	3.94 [3.86; 4.03]	4.05 [3.91; 4.32]	3.75 [3.48; 3.97]	
SB, c.u.	NO COPD	0.528 [0.494; 0.615]	0.567 [0.500; 0.671]	0.570 [0.505; 0.641]	10.69, 0.4698
	COPD I	0.533 [0.467; 0.642]	0.581 [0.487; 0.686]	0.508 [0.483; 0.548]	
	COPD II	0.533 [0.491; 0.677]	0.611 [0.561; 0.666]	0.536 [0.457; 0.652]	
MDA, μmol/L	NO COPD	7.35 [6.45; 10.85]	7.01 [5.90; 9.32]	6.75 [5.85; 8.97]	13.22, 0.2792
	COPD I	6.62 [5.64; 9.57]	6.45 [5.56; 8.89]	5.64 [5.13; 7.01]	
	COPD II	9.57 [5.98; 14.53]	9.79 [7.09; 11.62]	7.99 [5.60; 10.30]	

Note. n—the number of patients in the NO COPD/COPD I/COPD II groups, respectively; * differences between all groups are statistically significant (*p* ˂ 0.05). Na/K—sodium to potassium ratio, AST/ALT—aspartate aminotransferase to alanine aminotransferase ratio, LDH—lactatedehydrogenase, GGT—gamma glutamyltransferase, SOD—superoxide dismutase, AOA—antioxidant activity, DC—diene conjugates, SB—Schiff bases, MDA—malondialdehyde.

**Table 4 metabolites-11-00289-t004:** The structure of the study group.

Feature		Lung Cancer, n (%)		
		ADC, n = 189	SCC, n = 135	NEC, n = 68
**Age, years**		61.0 [56.0; 65.0]	59.0 [55.0; 66.5]	55.0 [52.0; 60.0]
**Gender**				
**Male**		129 (68.3)	128 (94.8)	50 (73.5)
**Female**		60 (31.7)	7 (5.2)	18 (26.5)
**Stage**				
**St IA**		16 (8.5)	3 (2.2)	5 (7.4)
**St IB**		52 (27.5)	28 (20.7)	10 (14.7)
**St IIA+B**		23 (12.2)	19 (14.1)	6 (8.8)
**St IIIA**		25 (13.2)	34 (25.2)	10 (14.7)
**St IIIB**		17 (9.0)	24 (17.8)	17 (25.0)
**St IV**		56 (29.6)	27 (20.0)	20 (29.4)
**COPD**				
**No**		113 (59.8)	69 (51.1)	41 (60.3)
**Yes**	GOLD I	57 (30.2)	40 (29.6)	17 (25.0)
	GOLD II	18 (9.5)	23 (17.1)	10 (14.7)
	GOLD III	1 (0.5)	3 (2.2)	-
**Smoking**				
**No**		112 (59.3)	64 (47.4)	43 (63.2)
**Yes**	Smokers I	14 (7.4)	4 (3.0)	5 (7.4)
	Smokers II	18 (9.5)	26 (19.3)	11 (16.2)
	Smokers III	45 (23.8)	41 (30.3)	9 (13.2)

Note. ADC—adenocarcinoma, SCC—squamous cell carcinoma, NEC—neuroendocrine cancer.

## Data Availability

The data presented in this study are available on request from the corresponding author. The data are not publicly available because they are required for the preparation of a Ph.D. thesis.

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
