# Peer review of "Features of the Metabolic Profile of Saliva in Lung Cancer and COPD: The Effect of Smoking Status"

_metabolites, 2021, doi:10.3390/metabo11050289_

Round 1
Reviewer 1 Report
The authors have addresses my initial comments regarding the original manuscript. I am therefore happy for the manuscript to be accepted.
Reviewer 2 Report
I have no further comments regarding the revised manuscript.
This manuscript is a resubmission of an earlier submission. The following is a list of the peer review reports and author responses from that submission.
Round 1
Reviewer 1 Report
This manuscript examines the metabolite profiles in the saliva of lung cancer and COPD non-smoking and smoking patients. This manuscript is well written and is novel. The findings are potentially important. There are only minor changes/comments that need to be addressed mainly in the discussion. These are outlined below:- The authors should outline the most significant differences in metabolites and comment on how these metabolites could help develop/promote COPD and/or lung cancer.
- The authors should comment further on how the findings of differential profiles could influence clinical practice in terms of diagnosis/prognosis/treatment.
- Was there any follow-up saliva's processed/analysed in this study or other previous studies (i.e.: tracing the same patient longitudinally over a period of time through their disease? The authors should comment on this as these studies may be important.
Reviewer 2 Report
Review : Features of the salivary metabolome in lung cancer and COPD: the effect of smoking status
General comments : The manuscript reports salivary biochemical composition of lung cancer patients regarding to their smoking status and COPD status. A number of biochemical traits such as the pH, mineral composition, some elements of protein metabolism, activity of enzymes (ALT, AST, ALP, LDH, CGT ..), lipoperoxidation products and the activity of antioxidant enzymes were determined. However this study is very descriptive where a ‘real’ hypothesis is missing and the results do not support the fact that the metabolic changes in saliva occur in lung cancer in combination with COPD, depending on the smoking factor. The statistics, although superfluous is inadequate. This type of experiment involving several groups should be analysed with several statistics approaches such as ANOVA and appropriate multivariate analysis like PLS-DA (Partial Least square discriminant analysis) where biochemical features contributing most to variation or separation are identified for further analysis. PCA is very useful for explorative analysis of the data, for example, to detect outliers. Therefore, initial application of PCA provides an informative first look at the dataset structure and relationships between groups. Ideally, the results of PCA analyses should be used to formulate an initial biological conclusion, which PLS can then verify and test in more detail. The principal reason for this is due to the fact that separation is only observed between groups in PCA scores when within-group variation is significantly less than between-group variation in the data, while separation in PLS scores may simply be fortuitous. It was shown that unsupervised (PCA) and supervised (PLS-DA) methods are very useful to assist in the biological interpretation of the data. Therefore, the application of a supervised analysis method is always followed by validation steps to make sure that the observed differences are significant and generalize well to new data.
In this article, there is also a lack of a comprehensive description of the salivary metabolome. Indeed the data presented here are interesting but do not explain how the salivary metabolism is involved in lung cancer in combination with COPD and why salivary biomarkers are relevant in that case.
The salivary metabolism changes should be explained and should be linked to the smoking factor and the presence of COPD. Why the saliva is suitable for studying metabolic processes in lung cancer? The discussion is just a comparison and a description of observation.
The results are very poor, since the statistical analysis is not appropriate and the effect of smoking status impact on the salivary metabolome in lung cancer and COPD is poorly demonstrated. Thus, I don’t recommend this feature article for publication in Metabolites.
Detailed Comments
-Introduction
If the exact mechanism of the appearance of markers of distal pathologies in saliva has not been yet been determined, how could this study explain that the composition of saliva can confirm its potential applicability for diagnostic purposes? Please leave a clear message, why to be interested in these metabolic characteristics according to 34 chosen indicators?
-Results
- 2.1: The number of patients in some groups are small, inducing some difficulties for group comparisons
-Please correct p5 line 148 2.4 instead of 3.4
The statistical analysis needs to be done with appropriate methods (see above)
-Discussion
The discussion fails to highlight the importance of this study and is just a reiteration of the results
Reviewer 3 Report
Features of the salivary metabolome in lung cancer and COPD: The effect of smoking status
Authors
Lyudmila V. Bel’skaya et al.
Manuscript Nr.: metabolites-1158368
In the present study, the biochemical analysis of saliva was conducted to differentiate between participants having smoking history, lung cancer and chronic obstructive pulmonary disease. Since the method of gaining the biological samples is non-invasive it provides an interesting option to perform the projected biochemical comparison. The potentials of using saliva for diagnostic purposes (see also “Saliva diagnostics – Current views and directions” https://doi.org/10.1177/1535370216681550) has gained much attention in this respect, whereby the sample collection is simple, but not mundane either. The introduction gives the necessary background why this approach was carried out, being short and compact. On the other end, it could be more comprehensive while citing other studies (lines 48-49) and discussing the factors taken into consideration e.g. while determining lung cancer. Further, it would also be more helpful for the reader to give some few examples of the relevant changes of biochemical indicators (line 53) as previously shown. The authors have described in a previous paper the methods they also use in this paper. However, why fewer parameters are used in the current work is not so understandable. This approach needs to be substantiated. The presentation of the results is rather descriptive and the quality of the figures could be improved. The parameters used to evaluate them statistically in the present study are standard options available for saliva analysis. These parameters are however subject to strong fluctuations which can be caused among other things also by the accompanying pharmacotherapy, whereby of course smoking also plays a determining role. In the case of smoker status, the question remains open as to whether they are still active and what the current abuse is or was. (This is detectable as reported in “AHRR methylation predicts smoking status and smoking intensity in both saliva and blood DNA” https://doi.org/10.1002/ajmg.b.32760). Further, it would then be helpful to analyse how the parameters used change in smokers who have neither COPD nor lung cancer as compared to healthy non-smokers. It can be assumed that COPD patients are treated and a corticoid therapy plays a role, this would also have to be taken into account. This was not done or no indication of it was given, thereby statistically not considered correlations may arise. These aspects need to be taken into the discussion, especially since this makes the groups even more inhomogeneous. The authors report significant changes in selected electrolyte components of saliva while comparing the different groups and considering smoking and COPD status, what does this information reveal? Why is it so – what metabolic mechanisms underlie these observations - some discussion to this respect would also be helpful. Therefore, altogether a solid and an interesting study, but it does have some weaknesses which need to be integrated in the discussion.
Specific remarks:
The title refers to “salivary metabolome” which is actually misleading since it is a more comprehensive biochemical evaluation of the registered clinical parameters. No “Omics” technologies were applied to have a more comprehensive metabolic data basis.
The parameters used for the statistics are subject to strong fluctuations even under therapy for COPD.
Detection of whether patients are active smokers is possible by simple bioanalysis, but is not provided. Thus, the correlation between smoking and the used values are only assumptions but not relevant.
The article is not suitable for publication as it is.
Reviewer 4 Report
This manuscript seems to be interesting, however, there are several problems to be solved.
#1. The authors investigated the salivary metabolome in lung cancer and COPD. The clinical implications of present study seem to be unclear to me. The authors should comment on this. #2. The authors reported the results of pathological classification of studied patients. The first concern of mine was whether the salivary metabolome is different in the different pathology of lung cancer. Furthermore, the second concern was whether the salivary metabolome was different according to the aging, or to the presence or absence of lung cancers. The authors should provide such information.